# Association between gestational period and obesity in children with the use of dermatoglyphic traits: A preliminary study

Adriano Alberti[1][ORCID]*, Jefferson Traebert[1], Eliane Traebert[1], Rudy José Nodari Junior[2], Clarissa Martinelli Comim[1]

1 Postgraduate Program in Health Sciences, Southern University of Santa Catarina (PPGCS-UNISUL), Palhoça, Santa Catarina, Brazil, 2 Postgraduate program in Biosciences and Health at University of the West of Santa Catarina, Joaçaba, Santa Catarina, Brazil

⊙ These authors contributed equally to this work.
* adrianoalberti90@hotmail.com

**Data Availability Statement:** All relevant data are within the paper and its Supporting Information files.

## Abstract

Events occurring during the gestational period can influence the development of diseases and conditions such as obesity. This study aimed to analyze the association between events occurring in the gestational period and the occurrence of obesity in children based on dermatoglyphic traits. The sample comprised 73 children born in 2009, living in Palhoça, Santa Catarina (SC), Brazil, regularly enrolled in public and private schools in that municipality and who are participants of an ongoing major cohort study project called Coorte Brasil Sul. The results show predictive traits of obesity when comparing BMI and fingerprint groups. Obese male group, presented the figure Ulnar Loop (UL) in the right hand (MDT1) thumb and greater number of ridges in the (MDSQL1) right thumb the greater the BMI; likewise, the older the woman getting pregnant the greater the number of ridges that the child presented in the (MESQL2) left index finger and (MESQL1) right thumb. The results obtained infer the presence of predictive traits of BMI ranges and a possible association between the dermatoglyphic traits of children with obesity and late pregnancy women.

## 1 Introduction

The maintenance of health conditions during the gestational period directly influences the healthy development of the fetus. Studies show that events that occur in this phase can directly influence the child's life and development [1, 2].Thus, it is essential to care for pregnant women with regard to habits that are harmful to their health, such as the use of alcohol and tobacco, among other legal and illegal substances, which are harmful [3]. In addition to preventing harmful habits, it is essential to practice healthy habits for a healthy pregnancy, such as good nutrition and prenatal care [4, 5]. Mainly in case of late pregnancy, that is, pregnancies in women after they are 35 years of age [6], more care is required, since pregnant women are at greater risk of miscarriage, premature birth, fetal malformation, development of

**Funding:** The author(s) received no specific funding for this work.

**Competing interests:** The authors have declared that no competing interests exist.

hypertension and gestational diabetes [6, 7]. Gestational diabetes increases the risk of developing obesity and metabolic disorders in childhood [8, 9].

Human fetal development begins on the 9th week of pregnancy and extends to childbirth [10]. Events that occurred during this period can affect development, causing malformation of the fetus [11]. Epidemiological and experimental studies have shown that some diseases can have their origin in fetal life [8]. And the intrauterine environment can induce epigenetic changes and these changes can cause the child to present with a predisposition to the development of metabolic diseases acquired in the uterus that are associated with obesity [12, 13].

In this connection, some traits can develop during this period, such as dermatoglyphics which is a method of observing fingerprints as a mark of fetal development [14–16]. Papillary designs become established between the 12th and 24th week of fetal life, following the development and maturation of the Central Nervous System (CNS), and remain unchanged throughout life [14, 17]. Fetal programming of diseases and conditions can originate during this period [18]. In this connection, studies have been carried out in order to identify dermatoglyphic traits that can be associated with different types of diseases and conditions [19, 20], including obesity [21, 22].

Obesity is the second cause of death that can be prevented and is associated with chronic health problems, such as diabetes and high blood pressure, among other diseases that demand high public health expenditure [23–25]. A study carried out by Alberti et al. [22], with 2,172 adolescents of both genders, identified dermatoglyphic traits predictive of obesity. In this connection, evidence has shown an association between cases of obesity and prenatal conditions and postnatal environmental factors, such as inadequate diet and physical inactivity [26, 27].

Dermatoglyphics is a non-invasive and low-cost method. To read fingerprints, the Gold Standard is the Dermatoglyphic Reader®, validated by Nodari Júnior et al. [28], as it is more accurate, practical and trustworthy when compared to the traditional dermatoglyphic method of paper ink and magnifying glass [28]. The identification of a dermatoglyphic mark that predicts obesity in children, associated with gestational age, may allow for more research and better referrals aimed at the prevention and/or adequate treatment of this condition throughout life, especially during childhood. In addition a similar research has never been carried out, which makes this study a major milestone in dermatoglyphics and innovation in health. This study aims to analyze the association between dermatoglyphics as a predictive marker of obesity in children and gestational age.

## 2 Method

### 2.1 Sample

We investigated children of both genders, born in 2009 (10–11 years old), residing in Palhoça, Santa Catarina (SC), regularly enrolled in public and private schools in the municipality; we decided to investigate this sample since these children participate in a larger cohort study project in progress called the South Brazil Cohort, under the Graduate Program in Health Sciences (PPGCS) of the *Universidade do Sul de Santa Catarina* (Unisul). All children participating in this project were born in 2009.

The sample size was calculated using the OpenEpi (Open Source Epidemiologic Statistics for Public Health) 3.03a program of the Rollins School of Public Health, Emory University, Atlanta, USA. The minimum number of individuals in the sample was determined using the following parameters: total population 1,756 children; 95% confidence level; unknown prevalence of the outcomes assessed (P = 50%). Thus, the total sample established included 316 children. To cover potential losses during the investigation, 10% of the number of children in the sample was added, totaling 347 children. It was collected from 360 children; however, due to

the exclusion criteria, the final sample totaled 73 children, according to Fig 1, since 184 children were excluded due to anomalous fingerprints or lack of temporal conditions, 94 for incomplete gestational data and 9 for whom no anthropometric data were available; In addition, it was not possible to collect additional data due to the Covid-19 pandemic, consequently, the collections process was closed.

Inclusion criteria: children who met the following joint criteria: born in 2009, enrolled in schools in the municipality (both public and private schools) residing in Palhoça, SC, who had had their BMI, dermatoglyphics and their mothers' gestational period data collected. Children who refused to have dermatoglyphic data (fingerprints) collected or who refused to carry out any of the anthropometric data measurements were excluded from the investigation. Children with anomalous fingerprints or without temporal collection conditions were also excluded. It was not possible to collect data from children who did not attend school on the days of data collection. Refusal to sign the Free and Informed Consent Form (FICF) by parents or guardians was also considered exclusion criterion.

The study was submitted to and approved by the Research Ethics Committee of the University of the South of Santa Catarina (Unisul), campus Palhoça, SC, Brazil, under opinion 3.362.267, according to the ethical standards of the rules and guidelines that govern research involving human beings, according to Resolution 466, 2012, of the National Health Council and to the Declaration of Helsinki.

## 2.2 Procedures

The anthropometric assessment was carried out as follows: weight was measured by a single measurement on a calibrated G-Tech® brand Glass 10 model digital scale, having a maximum capacity of 150 kg (Kg) and variation of ± 100 grams (g). The scale was placed on a flat, firm and smooth surface. The child was positioned in the center of the equipment, barefoot and with as little clothing as possible, upright, with feet together and arms extended alongside the body and looking at the horizon line, so that the body weight would evenly be distributed on both feet [28]. After the balance stabilized, the weight was read and, subsequently, the data were entered in a card.

To measure the children's height, a portable stadiometer Avanutri® (*Avanutri Equipamentos de Avaliação Ltda*, *Três Rios*, *RJ*) brand, was used having a measurement range from 20 cm to 200 cm and with a precision of 1 mm over the entire length. The child was positioned in the

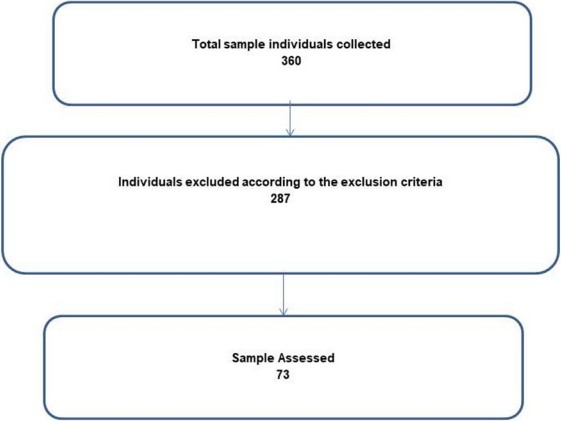

**Fig 1. Assessed sample flowchart.**

center of the equipment, fasting, barefoot and with no top head accessories, standing upright, with the arms extended along the body, head up, looking at a fixed point at eye level. The children's head was positioned in the Frankfurt plane (plane passing in the same horizontal line through the inferior margin of the left orbit and the upper margin of the external auditory meatus). The legs were positioned in parallel, the feet forming a right angle with the legs while heels, calves, buttocks, scapulae and upper part of the head (occipital region) were maintained in contact with the equipment; the sliding part of the equipment was then lowered until touching the child's head, maintaining enough pressure to compress his/her hair, then the measurement was taken [28].

The anthropometric assessment was performed based on the Body Mass Index (BMI), obtained by calculating the weight (kg) over height (meters) squared, according to the WHO standards. In addition to the index itself, the child's age and gender were also considered. The WHO defines obesity when the BMI is located on the curve above the score value z +2 for children over 5 years of age, as can be seen in Table 1.

The protocol proposed by Cummins and Midlo [15] was chosen to review the dermatoglyphic characteristics Fig 2. Drawings found by dermatoglyphic. For the capture, processing and analysis of fingerprints, the computerized process for dermatoglyphic reading was used. The reader consists of a rolling optical scanner, which collects and interprets the image and builds, in binary code, a drawing that is captured by a specific software for processing and reconstructing actual and binary images in black and white, through the Dermatoglyphic Reader® validated by Nodari Júnior and collaborators [28].

After the images were collected, the Dermatoglyphic Reader® user selected them one by one to define the points (nucleus and deltas), automatically drawing the Galton Line, so that the software, through specific algorithms, could make the intersection of the line drawn with the ridges of the fingerprint, thus providing the number of ridges for each finger, as well as the type of design of each fingerprint. The software makes the qualitative identification of the image and measures the number of ridges, generating the computerized spreadsheet resulting from the processed data [29].

The documentary data refer to the prenatal period of the sample, collected in 2015, through the study *Coorte Brasil Sul*. The documentary evaluation was carried out by reviewing the data obtained in the questionnaire of the study *Coorte Brasil Sul*. The data were collected in 2015 and the questionnaire was developed by the participants of the Brasil Sul Cohort. It contains several questions regarding the first 1,000 days of children's lives. It is split into sections and, although the entire questionnaire was applied, the section used in this study which refers to the gestational events that were assessed in this investigation and were obtained from the mothers' answers was considered; drug use during pregnancy, child gender, report of the following ailment during pregnancy: chickenpox/varicella, cytomegalovirus, toxoplasmosis or cat

**Table 1. BMI cutoff points for children over 5 years of age.**

| Critical Values | Nutritional Diagnostic |
|---|---|
| < Score-z -2 | Low BMI for age |
| ≥Score-z -2 e | Adequate or eutrophic BMI |
| < Score-z +1 | |
| ≥ Score-z +1 e | Overweight |
| < Score-z +2 | |
| ≥ Score-z +2 | Obesity |

Source: Brasil [22].

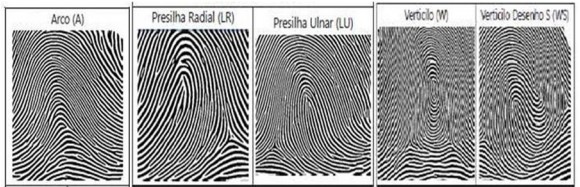

**Fig 2.**

disease, measles, rubella, syphilis, tetanus, pneumonia, diabetes, hypertension, heart disease were reviewed as well as alcohol intake, smoking and prenatal care conditions in addition to the pregnant women's age.

## 2.3 Statistical analysis

Descriptive data are presented as mean ± standard deviation, as well as absolute and percentage values (%) as appropriate. To determine whether the data were well modeled on a normal distribution, the Kolmogorov-Smirnov test was applied, followed by visual inspection of the charts. For statistical analysis of a few parameters, the values were standardized (by conversion to Z-score), in order to reduce the bias caused by data error without normal distribution. Correlations between continuous variables (Left hand, sum of the number of ridges of finger 1—thumb (MESQL1), left hand, sum of the number of ridges of finger 2—index (MESQL2), left hand, sum of the number of ridges of finger 3—middle finger (MESQL3), left hand, sum of the number of ridges of finger 4—ring (MESQL4) and left hand, sum of the number of ridges of finger 5 –little finger (MESQL5); sum of the total number of ridges of the left hand (SQTLE); right hand, sum of the number of ridges of finger 1—thumb (MDSQL1), right hand, sum of the number of ridges of finger 2—index (MDSQL2), right hand, sum of the number of ridges of finger 3—middle finger (MDSQL3), right hand, sum of the number of ridges of finger 4—ring (MDSQL4) and right hand, sum of the amount of ridges of finger 5 –little finger (MDSQL5); sum of the total number of ridges of the right hand (SQTLD); sum of the total number of ridges—both hands (LQTS), age at which the mother became pregnant, and BMI child, were assessed using Spearman's coefficient. To compare the frequencies observed in each interaction mediated by the presence of multiple nominal variables Arch (A), Radial Loop (RL), Ulnar Loop (UL), Whorl (W) and S Design, the chi-square test ($\chi^2$) was applied, and additionally the value of Fisher's exact test was assessed. Although it seems more appropriate for the study, we did not conduct multivariate analysis of variance (MANOVA), due to assumption violations. A multiple linear regression analysis was performed instead. Additionally, the hypothetical prediction of the child's BMI values from the dermatoglyphic profile data was explored through a multiple linear regression analysis. Those parameters of the dermatoglyphic profile (number of fingerprints of each finger) that did not contribute to the adjusted model were removed by the backward stepwise process. The coefficient for each variable was obtained from the adjusted model with the lowest probability value (p). All statistical analyses were performed using the Statistical Package for Social Sciences program (SPSS–IBM Statistics, version 21).

## 3 Results

The results obtained demonstrate that there was a significant correlation between the number of ridges on the child's fingers, BMI child and maternal age at the beginning of pregnancy, with a greater number of ridges on fingers MESQL1 (left thumb) and MDSQL1 (right thumb),

when considering both genders. This means that the greater the number of ridges the higher the BMI child; also, the lower the number of ridges the lower the BMI child, as shown in Table 2. Moreover, Tables 2, 3 and 4 show that the number of ridges on the child's MESQL2 (left index) and MDSQL1 (right thumb) increased with increasing maternal age at pregnancy onset.

In Table 3, the sample was separated in relation to gender (boys and girls analyzed separately).It was observed that this correlation between finger ridges and BMI child remained in the MDSQL1 finger (right thumb) in male individuals.

In Table 4, the sample is separated according to gender (boys and girls analyzed separately). There was no significant correlation in the female individuals.

Tables 5 and 6 show the relationship between BMI child and patterns of fingerprints, according to gender. In the MDT1 finger there was a relationship between BMI child and the fingerprint pattern in male individuals.

The results obtained indicate the presence of predictive markers of BMI child range on the MDT1 finger. The low weight group had a higher frequency of the ulnar loop (UL) pattern (41.7%), followed by the arch (A) pattern (33.3%); the normal weight group exhibited a higher frequency of the ulnar loop (UL) pattern (47.4%), followed by the whorl (W) pattern (42.1%); the pre-obese group had a higher frequency of the S pattern; and the obese group presented with a higher frequency of the ulnar loop (UL) pattern.

In Table 7, after removing the interaction between Gestational Health and Finger ridges of all fingers, there is no significant effect that explains the changes observed in the dependent variable (BMI) child.

## 4 Discussion

This study showed predictive obesity traits when BMI child and fingerprint groups were correlated. When the sample was analyzed considering both genders, the higher the BMI child the greater the number of ridges in MESQL1 (left thumb) and MDSQL1 (right thumb). However,

**Table 2. Correlation between number of ridges on the fingers, BMI child, and age at beginning of pregnancy of the mother (AGM age) both genders.**

|  | BMI Child | Age AGE |
|---|---|---|
| MESQL1 | .254* | -.005 |
| MESQL2 | .030 | .271* |
| MESQL3 | .109 | .102 |
| MESQL4 | .126 | .163 |
| MESQL5 | .159 | .112 |
| SQTLE | .162 | .152 |
| MDSQL1 | .254* | .272* |
| MDSQL2 | -.036 | .114 |
| MDSQL3 | .028 | .108 |
| MDSQL4 | .028 | .115 |
| MDSQL5 | .160 | .056 |
| SQTLD | .181 | .180 |
| SQTL | .192 | .181 |

Table present values of Spearman correlation coefficient.

* Significant at level of $p < 0.05$.

Source: Alberti et al. [16]

**Table 3. Correlation between number of ridges on the fingers, BMI child, and age (AGM age) male (at onset of pregnancy).**

|  | BMI Child | Age AGE |
|---|---|---|
| MESQL1 | .246 | -.005 |
| MESQL2 | .026 | .271* |
| MESQL3 | .151 | .102 |
| MESQL4 | .175 | .163 |
| MESQL5 | .075 | .112 |
| SQTLE | .193 | .152 |
| MDSQL1 | .354* | .272* |
| MDSQL2 | -.066 | .114 |
| MDSQL3 | -.058 | .108 |
| MDSQL4 | .142 | .115 |
| MDSQL5 | .046 | .056 |
| SQTLD | .270 | .180 |
| SQTL | .282 | .181 |

Table present values of Spearman correlation coefficient.

* Significant at level of $p<0.05$.

Source: Alberti et al. [16]

considering that men and women have different dermatoglyphic patterns [30, 31], when the sample was analyzed according to the gender only the male individuals showed a significant difference in the MDSQL1 finger (right thumb), i.e., the greater the number of ridges, the greater the BMI. There was a relationship between BMI and MDT1 fingerprint pattern in male individuals, with a higher frequency of the ulnar loop (UL) pattern in the obese group. The analysis of the association between BMI, fingerprints and gestational data showed no significant differences; however, the analysis of the correlation between the number of finger ridges,

**Table 4. Correlation between number of ridges on the fingers, BMI child, and age of onset of pregnancy of the mother (AGM age) female.**

|  | BMI Child | Age AGE |
|---|---|---|
| MESQL1 | .263 | -.005 |
| MESQL2 | .060 | .271* |
| MESQL3 | .107 | .102 |
| MESQL4 | .092 | .163 |
| MESQL5 | .184 | .112 |
| SQTLE | .169 | .152 |
| MDSQL1 | .140 | .272* |
| MDSQL2 | -.019 | .114 |
| MDSQL3 | .106 | .108 |
| MDSQL4 | -.064 | .115 |
| MDSQL5 | .305 | .056 |
| SQTLD | .129 | .180 |
| SQTL | .150 | .181 |

Table present values of Spearman correlation coefficient.

* Significant at level of $p<0.05$.

Source: Alberti et al. [16]

**Table 5. Relationship between BMI child and digital types.**

| Female | Digital Brand | Low Weight | | Normal Weight | | Pre-Obese | | Obese | | $\chi^2$ |
|---|---|---|---|---|---|---|---|---|---|---|
| | | *n* | % | *n* | % | *N* | % | *n* | % | |
| MET1 | A | 2 | 10.5 | 4 | 30.8 | 0 | 0 | 0 | 0 | .812 |
| | LR | 1 | 5.3 | 0 | 0 | 0 | 0 | 0 | 0 | |
| | LU | 10 | 52.6 | 4 | 30.8 | 3 | 42.9 | 1 | 100 | |
| | W | 3 | 15.8 | 3 | 23.1 | 2 | 28.6 | 0 | 0 | |
| | S | 3 | 15.8 | 2 | 15.4 | 2 | 28.6 | 0 | 0 | |
| MET2 | A | 4 | 21 | 1 | 7.7 | 1 | 14.3 | 0 | 0 | |
| | LR | 4 | 21 | 3 | 23.1 | 1 | 14.3 | 0 | 0 | |
| | LU | 4 | 21 | 5 | 38.5 | 1 | 14.3 | 0 | 0 | .929 |
| | W | 6 | 31.6 | 3 | 23.1 | 3 | 42.6 | 1 | 100 | |
| | S | 1 | 5.3 | 1 | 7.7 | 1 | 14.3 | 0 | 0 | |
| MET3 | A | 1 | 5.3 | 0 | 0 | 1 | 14.3 | 0 | 0 | |
| | LR | 0 | 0 | 1 | 7.7 | 0 | 0 | 0 | 0 | |
| | LU | 14 | 73.7 | 6 | 46.2 | 5 | 71.4 | 1 | 100 | .696 |
| | W | 3 | 15.8 | 3 | 23.1 | 1 | 14.3 | 0 | 0 | |
| | S | 1 | 5.3 | 3 | 23.1 | 0 | 0 | 0 | 0 | |
| MET4 | A | 0 | 0 | 1 | 7.7 | 1 | 14.3 | 0 | 0 | |
| | LR | 0 | 0 | 0 | 0 | 0 | 0 | 0 | 0 | |
| | LU | 14 | 73.7 | 7 | 53.8 | 2 | 28.6 | 1 | 100 | .544 |
| | W | 4 | 21 | 5 | 38.4 | 3 | 42.9 | 0 | 0 | |
| | S | 1 | 5.3 | 0 | 0 | 1 | 14.3 | 0 | 0 | |
| MET5 | A | 1 | 5.3 | 1 | 7.7 | 2 | 28.6 | 0 | 0 | |
| | LR | 0 | 0 | 0 | 0 | 0 | 0 | 0 | 0 | |
| | LU | 13 | 68.4 | 11 | 84.6 | 4 | 57.1 | 1 | 100 | .491 |
| | W | 5 | 26.3 | 1 | 7.7 | 1 | 14.3 | 0 | 0 | |
| | S | 0 | 0 | 0 | 0 | 0 | 0 | 0 | 0 | |
| MDT1 | A | 1 | 5.3 | 2 | 15.4 | 2 | 28.6 | 0 | 0 | |
| | LR | 0 | 0 | 0 | 0 | 0 | 0 | 0 | 0 | |
| | LU | 9 | 47.4 | 5 | 38.5 | 2 | 28.6 | 1 | 100 | .881 |
| | W | 4 | 21 | 2 | 15.4 | 1 | 14.3 | 0 | 0 | |
| | S | 5 | 26.3 | 4 | 30.8 | 2 | 28.6 | 0 | 0 | |
| MDT2 | A | 0 | 0 | 5 | 38.5 | 2 | 28.6 | 0 | 0 | |
| | LR | 5 | 26.3 | 3 | 23 | 0 | 0 | 0 | 0 | |
| | LU | 7 | 36.8 | 1 | 7.7 | 2 | 28.6 | 1 | 100 | .258 |
| | W | 5 | 26.3 | 2 | 15.4 | 2 | 28.6 | 0 | 0 | |
| | S | 2 | 10.5 | 2 | 15.4 | 1 | 14.3 | 0 | 0 | |
| MDT3 | A | 0 | 0 | 2 | 15.4 | 1 | 14.3 | 0 | 0 | |
| | LR | 0 | 0 | 0 | 0 | 0 | 0 | 0 | 0 | |
| | LU | 13 | 68.4 | 8 | 61.5 | 5 | 71.4 | 1 | 100 | .317 |
| | W | 5 | 26.3 | 0 | 0 | 1 | 14.3 | 0 | 0 | |
| | S | 1 | 5.3 | 3 | 23 | 0 | 0 | 0 | 0 | |
| MDT4 | A | 0 | 0 | 4 | 30.8 | 0 | 0 | 0 | 0 | |
| | LR | 0 | 0 | 0 | 0 | 0 | 0 | 0 | 0 | |
| | LU | 12 | 63.2 | 3 | 23 | 3 | 42.9 | 1 | 100 | .145 |
| | W | 6 | 31.6 | 5 | 38.5 | 4 | 57.1 | 0 | 0 | |
| | S | 1 | 5.3 | 1 | 7.7 | 0 | 0 | 0 | 0 | |

(*Continued*)

**Table 5.** (Continued)

| Female | Digital Brand | Low Weight | | Normal Weight | | Pre-Obese | | Obese | | $\chi^2$ |
|--------|---------------|-----|------|-----|------|-----|------|-----|------|-----|
| | | n | % | n | % | N | % | n | % | |
| MDT5 | A | 3 | 15.8 | 2 | 15.4 | 1 | 14.3 | 0 | 0 | |
| | LR | 0 | 0 | 0 | 0 | 0 | 0 | 0 | 0 | |
| | LU | 12 | 63.2 | 9 | 69.2 | 4 | 57.1 | 1 | 100 | .983 |
| | W | 4 | 21 | 2 | 15.4 | 2 | 28.6 | 0 | 0 | |
| | S | 0 | 0 | 0 | 0 | 0 | 0 | 0 | 0 | |

Source: Alberti et al.[16].

BMI, and maternal age at the beginning of pregnancy showed that the number of ridges the child exhibited in the left index (MESQL2) and right thumb (MESQL1), which was the finger in which the highest number of ridges/higher BMI relationship was established in boys, increased with mothers' increasing age at pregnancy onset.

The most recent studies begin to understand obesity as a multifactorial condition, involving environmental, genetic and epigenetic aspects [32, 33]. The environmental factor can be controlled through daily behavioral changes involving energy intake and expenditure [33]. Genetic factors play a crucial role in determining an individual's predisposition to weight gain and obesity. Several genetic variants were identified as monogenic forms of human obesity, having success overrunning common polygenic forms. In the context of molecular genetics, the approach of genome-wide association studies (GWAS) and their findings showed a series of genetic variants that predispose to obesity [21, 34].

In addition to environmental and genetic factors, epigenetics appears to be fundamental in the development of obesity, because potential mechanisms of epigenetic changes may be involved as mediators of environmental influences; these changes play an important role in weight gain [35]. Every organism is unique and has an epigenetic trait, inherited and generated in the womb. This has led to studies that aim to highlight the influence of the fetal period and environment in the development of diseases and conditions throughout an individual's life, such as obesity [36].

Dermatoglyphics has its fundamental basis in this premise, since it is an epigenetic marker associated with the period of fetal development [37]. In the present study, the analysis of the association between BMI, fingerprints and gestational data showed no significant results; however, it was observed that the older the woman when she became pregnant, the greater the number of ridges on the child's left index and right thumb. These findings relate to one of the predictive obesity traits presented in this study, which is the higher number of ridges on MDSQL1 (right thumb) in male individuals, i.e., there seems to be a relationship between a higher number of ridges, which is a predictive obesity trait, and late age pregnancy.

Alberti et al. [22] demonstrated a predictive obesity trait, with a higher total number of ridges in the MESQL2 left hand index finger, a higher frequency of the whorl (W) pattern in the healthy-weight group, a higher frequency of the radial loop (RL) pattern in the overweight group, and a higher frequency of the ulnar loop (UL) pattern in the obese group. These results are in line with the findings of the present study, which also indicated a greater number of ridges in children with obesity, namely in the sum of the right thumb MDSQL1 in male individuals, and the same fingerprint pattern in the group of children with obesity, i.e., the ulnar loop (UL) pattern. Despite the fact that the study by Alberti et al. [22] shows a greater number of ridges on a finger of the left hand and in the present study on a finger of the right hand, this

**Table 6. Relationship between BMI child and digital types.**

| Male | Digital Brand | Low Weight | | Normal Weight | | Pre-Obese | | Obese | | $\chi^2$ |
|---|---|---|---|---|---|---|---|---|---|---|
| | | n | % | n | % | N | % | n | % | |
| MET1 | A | 0 | 0 | 1 | 5.3 | 0 | 0 | 0 | 0 | .733 |
| | LR | 0 | 0 | 0 | 0 | 0 | 0 | 0 | 0 | |
| | LU | 8 | 66.7 | 8 | 42.1 | 1 | 100 | 1 | 100 | |
| | W | 3 | 25 | 3 | 15.8 | 0 | 0 | 0 | 0 | |
| | S | 1 | 8.3 | 7 | 36.8 | 0 | 0 | 0 | 0 | |
| MET2 | A | 0 | 0 | 0 | 0 | 0 | 0 | 0 | 0 | |
| | LR | 3 | 12 | 2 | 10.5 | 0 | 0 | 0 | 0 | |
| | LU | 5 | 41.7 | 8 | 42.1 | 0 | 0 | 1 | 100 | .400 |
| | W | 2 | 16.7 | 7 | 36.8 | 0 | 0 | 0 | 0 | |
| | S | 2 | 16.7 | 2 | 10.5 | 1 | 100 | 0 | 0 | |
| MET3 | A | 2 | 16.7 | 2 | 10.5 | 0 | 0 | 0 | 0 | |
| | LR | 0 | 0 | 1 | 5.3 | 0 | 0 | 0 | 0 | |
| | LU | 9 | 75 | 11 | 57.9 | 1 | 100 | 1 | 100 | .955 |
| | W | 0 | 0 | 4 | 21 | 0 | 0 | 0 | 0 | |
| | S | 1 | 8.4 | 1 | 5.3 | 0 | 0 | 0 | 0 | |
| MET4 | A | 2 | 16.7 | 0 | 0 | 0 | 0 | 0 | 0 | |
| | LR | 0 | 0 | 1 | 5.3 | 0 | 0 | 0 | 0 | |
| | LU | 9 | 75 | 10 | 52.6 | 1 | 100 | 0 | 0 | .427 |
| | W | 1 | 8.4 | 5 | 26.3 | 0 | 0 | 1 | 100 | |
| | S | 0 | 0 | 3 | 15.8 | 0 | 0 | 0 | 0 | |
| MET5 | A | 1 | 8.3 | 0 | 0 | 0 | 0 | 0 | 0 | |
| | LR | 0 | 0 | 0 | 0 | 0 | 0 | 0 | 0 | |
| | LU | 11 | 91.7 | 15 | 78.9 | 1 | 100 | 1 | 100 | .843 |
| | W | 0 | 0 | 3 | 15.8 | 0 | 0 | 0 | 0 | |
| | S | 0 | 0 | 1 | 5.3 | 0 | 0 | 0 | 0 | |
| MDT1 | A | 4 | 33.3 | 0 | 0 | 0 | 0 | 0 | 0 | |
| | LR | 0 | 0 | 0 | 0 | 0 | 0 | 0 | 0 | |
| | LU | 5 | 41.7 | 9 | 47.4 | 0 | 0 | 1 | 100 | .048* |
| | W | 2 | 16.7 | 8 | 42.1 | 0 | 0 | 0 | 0 | |
| | S | 1 | 8.3 | 2 | 10.5 | 1 | 100 | 0 | 0 | |
| MDT2 | A | 1 | 8.3 | 2 | 10.5 | 0 | 0 | 0 | 0 | |
| | LR | 1 | 8.3 | 2 | 10.5 | 0 | 0 | 0 | 0 | |
| | LU | 4 | 33.3 | 5 | 26.3 | 0 | 0 | 1 | 100 | .859 |
| | W | 3 | 25 | 7 | 36.8 | 0 | 0 | 0 | 0 | |
| | S | 3 | 25 | 3 | 15.8 | 1 | 100 | 0 | 0 | |
| MDT3 | A | 0 | 0 | 1 | 5.3 | 0 | 0 | 0 | 0 | |
| | LR | 0 | 0 | 1 | 5.3 | 0 | 0 | 0 | 0 | |
| | LU | 8 | 66.6 | 9 | 47.4 | 0 | 0 | 1 | 100 | .065 |
| | W | 3 | 25 | 8 | 42.1 | 0 | 0 | 0 | 0 | |
| | S | 1 | 8.3 | 0 | 0 | 1 | 100 | 0 | 0 | |
| MDT4 | A | 1 | 8.3 | 1 | 5.3 | 0 | 0 | 0 | 0 | |
| | LR | 0 | 0 | 0 | 0 | 0 | 0 | 0 | 0 | |
| | LU | 7 | 58.3 | 6 | 31.7 | 0 | 0 | 0 | 0 | .097 |
| | W | 3 | 25 | 11 | 57.9 | 0 | 0 | 1 | 100 | |
| | S | 1 | 8.3 | 1 | 5.3 | 1 | 100 | 0 | 0 | |

*(Continued)*

**Table 6.** (Continued)

| Male | Digital Brand | Low Weight | | Normal Weight | | Pre-Obese | | Obese | | $\chi^2$ |
|---|---|---|---|---|---|---|---|---|---|---|
| | | *n* | % | *n* | % | *N* | % | *n* | % | |
| MDT5 | *A* | 1 | 8.3 | 0 | 0 | 0 | 0 | 0 | 0 | |
| | *LR* | 0 | 0 | 0 | 0 | 0 | 0 | 0 | 0 | |
| | *LU* | 9 | 75 | 14 | 73.7 | 1 | 100 | 1 | 100 | .929 |
| | *W* | 2 | 16.7 | 3 | 15.8 | 0 | 0 | 0 | 0 | |
| | *S* | 0 | 0 | 2 | 10.5 | 0 | 0 | 0 | 0 | |

Source: Alberti et al. [16]

evidence enhances the fact that obese children exhibit a greater number of finger ridges, since the method of both studies was similar.

Pediatric obesity has its basis in genetic susceptibilities influenced by a permissive environment starting *in utero* and extending throughout childhood and adolescence [38]. Comorbidities resulting from this condition are common and often result in long-term health complications. Obesity screening must be applied logically and correctly for early identification before more serious complications occur [38]. The results obtained in the present study and that of Alberti et al. [22] corroborated the fact that the higher number of lines in obese children indicates a dermatoglyphic mark that could serve as a basis for future obesity screening studies. There is strong experimental evidence indicating that specific epigenetic marks represent a kind of fingerprint of the intrauterine programming [13].

In another study [21], with a sample of 370 children with obesity, aiming to identify dermatoglyphic patterns in obese individuals and determine the association between dermatoglyphic patterns and obesity, the results showed a high frequency of the A pattern on the right thumb and a low number of ridges. However, in that study, the BMI reference criteria were different in relation to the present study, thus being able to influence the results and explain the difference found. In the present study the anthropometric assessment was used in accordance with the WHO recommendation, based on the Body Mass Index (BMI), obtained by calculating the

**Table 7. Regression table.**

| | Body Mass Index (BMI) child | | | |
|---|---|---|---|---|
| | $R^2$ | Coefficient β | *F* | *p* |
| MESQL1 | .061 | .441 | .402 | .941 |
| MESQL2 | | .077 | | |
| MESQL3 | | .555 | | |
| MESQL4 | | .151 | | |
| MESQL5 | | .229 | | |
| SQTLE | | -1.599 | | |
| MDSQL1 | | .510 | | |
| MDSQL2 | | .151 | | |
| MDSQL3 | | .054 | | |
| MDSQL4 | | .189 | | |
| MDSQL5 | | .327 | | |
| SQTLD | | -1.785 | | |
| SQTL | | -2.071 | | |

Source: Alberti et al. [16]

weight (kg) over height (meters) squared, considering, in addition to the index itself, the age and gender of the child according to the WHO standards [39], thus conveying greater confidence to the results obtained.

Another study [40] yielded similar results. It was sought to determine the dermatoglyphic characteristics of obese Ibibian patients by comparing a group of 50 obese individuals (25 men and 25 women) with a group of 50 normal weight subjects (25 men and 25 women), showing more frequent arches (A) on the first digits of the right hand in obese men (54.5%) and women (42.33%), whereas individuals with normal weight presented a higher frequency of ulnar loops (LU).

In a survey conducted by Pasetti; Gonçalves; Padovani [41] with 30 obese Brazilian women aged, on average, 46.1 ± 07.87 years, all with a BMI equal to or higher than 30%, showed that the participants had a predominance of the arch (A) pattern and a low frequency of the ulnar loop (UL) pattern. These results confirm the findings of other authors [21, 40] who also found a predominance of the arch (A) pattern in the obese group.

In other studies [40, 41] the samples were composed by adults, and it has been proven that obese children suffer more with genetic and epigenetic problems in the development of obesity than the obese adult [42]; therefore, the samples in these studies may have included adults who developed obesity not so much due to genetics, but more because of environmental conditions such as sedentary lifestyle and inadequate nutrition.

The results of other studies [21, 40, 41] were different from those obtained in the present study; however, the traditional method of collection and analysis with printer's ink and magnifying glass was used in these studies, which is not as accurate as using the *Leitor Dermatoglífico*® (Dermatoglyphic Reader) validated by Nodari Júnior et al. [29], the computerized method being four times more accurate than the traditional method, and the results obtained by Alberti et al. [22] who used the *Leitor Dermatoglífico*®, are in line with the findings of the present study. Another important factor is that the associations and correlations assessed in the present study were not included in the abovementioned studies. Therefore, the present study offers data that are rather relevant.

For future studies, the Dermatoglyphic Reader® validated by Nodari Júnior et al. should be used [22]. This device was used in the present study, and also by Alberti et al. [22]. On the other hand, the anthropometric assessment should also be performed following the WHO standards to obtain more reliable results that are better comparable to those of the present study.

The results show predictive obesity traits in children. In addition, the results were associated with gestational data and indicated that the older the woman was when she became pregnant, the greater the number of ridges in the child's MESQL2 (left index) and in the MDSQL1 (right thumb); the right thumb was the finger in which children with higher BMI exhibited a greater number of ridges. Data in the literature demonstrate that late pregnancy increases the risk for the development of several diseases and conditions throughout the child's life [43, 44], including childhood obesity [45].

The limitation of this study was the small sample of 73 children, since the total population calculated for the study was 316 children; however data collection was impaired due to the COVID-19 pandemic. The strong point is that, besides the identification of a dermatoglyphic mark that characterizes obese children since their birth using a method four times more accurate than the traditional one, this study is associated with maternal age; no other study has considered this link before. We conclude that children with obesity have a predictive dermatoglyphic obesity trait from birth and that mothers' late age pregnancy is a predictive factor of childhood obesity. It is expected that the data obtained in this study will contribute to develop an obesity prediction formula in children, using an easy and low cost method that should help to prevent and control this condition.

## 5 Conclusion

A predictive obesity trait was found in male children consisting of a greater number of finger ridges in the right thumb MDSQL1. As to the types of figures, the right thumb MDT1 exhibited a higher frequency of the Ulnar Loop (UL).

In addition to revealing a predictive obesity trait, the results show an association between dermatoglyphic traits in children with obesity and women who become pregnant at an older age; the older the woman at pregnancy onset the greater the number of ridges the obese male child exhibits on the right thumb,

## Supporting information

**S1 Data. Weight, dermatoglyphic and gestational data.**
(XLSX)

## Acknowledgments

I am grateful for the group's dedication.

## Author Contributions

**Conceptualization:** Adriano Alberti, Jefferson Traebert, Eliane Traebert, Rudy José Nodari Junior, Clarissa Martinelli Comim.

**Data curation:** Adriano Alberti, Rudy José Nodari Junior, Clarissa Martinelli Comim.

**Formal analysis:** Adriano Alberti, Jefferson Traebert, Eliane Traebert, Rudy José Nodari Junior, Clarissa Martinelli Comim.

**Funding acquisition:** Adriano Alberti, Jefferson Traebert, Eliane Traebert, Rudy José Nodari Junior, Clarissa Martinelli Comim.

**Investigation:** Adriano Alberti, Jefferson Traebert, Eliane Traebert, Rudy José Nodari Junior, Clarissa Martinelli Comim.

**Methodology:** Adriano Alberti, Jefferson Traebert, Eliane Traebert, Rudy José Nodari Junior, Clarissa Martinelli Comim.

**Project administration:** Adriano Alberti, Jefferson Traebert, Eliane Traebert, Rudy José Nodari Junior, Clarissa Martinelli Comim.

**Resources:** Adriano Alberti, Eliane Traebert, Rudy José Nodari Junior, Clarissa Martinelli Comim.

**Software:** Adriano Alberti, Rudy José Nodari Junior, Clarissa Martinelli Comim.

**Supervision:** Adriano Alberti, Jefferson Traebert, Rudy José Nodari Junior, Clarissa Martinelli Comim.

**Validation:** Adriano Alberti, Jefferson Traebert, Eliane Traebert, Rudy José Nodari Junior, Clarissa Martinelli Comim.

**Visualization:** Adriano Alberti, Jefferson Traebert, Eliane Traebert, Rudy José Nodari Junior, Clarissa Martinelli Comim.

**Writing – original draft:** Adriano Alberti, Clarissa Martinelli Comim.

**Writing – review & editing:** Adriano Alberti, Clarissa Martinelli Comim.

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
