## [Decision Letter · Decision Letter 0]

2 Aug 2021

PONE-D-21-09898

Association among events that occurred in the gestation period and obesity in children with the use of dermatoglyphic traits

PLOS ONE

Dear Dr. Alberti,

Thank you for submitting your manuscript to PLOS ONE. After careful consideration, we feel that it has merit but does not fully meet PLOS ONE’s publication criteria as it currently stands. Therefore, we invite you to submit a revised version of the manuscript that addresses the points raised during the review process.

We look forward to receiving your revised manuscript.

Kind regards,

Ramune Jacobsen

Academic Editor

PLOS ONE

Journal Requirements:

We will update your Data Availability statement to reflect the information you provide in your cover letter."

4. Please upload a new copy of Figure 2 as the detail is not clear. Please follow the link for more information: https://blogs.plos.org/plos/2019/06/looking-good-tips-for-creating-your-plos-figures-graphics/" https://blogs.plos.org/plos/2019/06/looking-good-tips-for-creating-your-plos-figures-graphics/.

Reviewers' comments:

Reviewer's Responses to Questions

**Comments to the Author**

1. Is the manuscript technically sound, and do the data support the conclusions?

Reviewer #1: Partly

Reviewer #2: Partly

Reviewer #3: No

2. Has the statistical analysis been performed appropriately and rigorously? 

Reviewer #1: Yes

Reviewer #2: No

Reviewer #3: N/A

3. Have the authors made all data underlying the findings in their manuscript fully available?

Reviewer #1: Yes

Reviewer #2: Yes

Reviewer #3: Yes

4. Is the manuscript presented in an intelligible fashion and written in standard English?

Reviewer #1: No

Reviewer #2: Yes

Reviewer #3: Yes

5. Review Comments to the Author

Reviewer #1: The paper presents interesting an unique study worth publication. However, there are some issues that need correction.

1. English language should be checked and corrected by proffesional English editor.

2. The introduction is too short- it should more deeply justify the need of the study and give more scientific background.

3. Please underline the novelty of the study in the introduction.

4. Justify the childrens' age. Why you have included children in the age of 10-11??

5. The total population calculated for the study was 316; however you managed to include only 73. This is great weak point of the study and should be pointed broadly in limitation. In title please point that this is a pilot study or preliminary study- the complete study should include the total of 316 children.

6. Provide the manufacturer of digitale scale. Was the body measured fasting?

7. In methods please clearly provide information on what kinds of "events that occurred in the gestation period" were analyzed in the study.

8. The RESULT section should start from the sentence "The results obtained demonstrate that there was a significant correlation between the number of ridges on the child’s fingers, ... " at line 216.

9. In table headings you use AGM abbrev., but in the table there is IGM- correct it.

10. In results (text and tables) clearly state whose BMI do you mean- child or maternal.

11. "Different genetic variants have been identified as monogenic forms of human obesity, having success over common polygenic forms." This sentence is not clear. Do you mean that monogenic forms of obesity predominate over polygenic? It is not true- correct it.

12. Your discussion is a simple enumaretion of the results of similar studies. Please provide some molecular/genetic mechanisms linking obesity and fingerprints, some clinical implications and directions of future studies in the issue.

13. In the study aim and title you pointed, that you want to analyze the association between events occurring in the gestational period and the occurrence of obesity in children from dermatoglyphic traits. However, in the study you did not present any results relating to "events occurring in the gestational period". Namely, you did not investigate events occurring in the gestational period. The only "event" analyzed was maternal age at gestation onset. Please correct study title and aim deleting "events occurring in the gestational period". Unfortunately, the lack of the analysis of "events occurring in the gestational period" significantly decreases the scientific value of the paper.

14. After limitations please add some study strong points.

15. "The study shows the application of research in dermatoglyphics as an epigenetic predictive obesity trait, and the results show predictive obesity traits in children. In addition, the results were associated with gestational data." Such sentence should be placed in the discussion section. Moreover, you did not discussed any "epigenetic" link between obesity and fingerprint. Thus, the use of word "epigenetic" in the paper is unjustifiable.

I recommand to accept the paper after major revision.

Reviewer #2: The authors have presented useful data

I have the following comments

1) The authors have used the term predictive traits; however, they have just used correlations. Even regressions have not been used for analysis

2) The results section merges into methods. It is difficult to understand the whole results section

3) Do we have information on the mothers? Nothing much has been included in the manuscript

Reviewer #3: Here the authors analyze the association between events during gestation and the occurrence of obesity in children from dermatoglyphic traits. The authors report a predictive traits of obesity when comparing BMI and fingerprint groups in the figure patterns for males.

1) This is a preliminary correlation study and does not have any novel mechanistic insight.

2) The sample size (73) is too small to have any predictive value.

6. PLOS authors have the option to publish the peer review history of their article (what does this mean?). If published, this will include your full peer review and any attached files.

Reviewer #1: No

Reviewer #2: No

Reviewer #3: No

---

## [Author Response · Author response to Decision Letter 0]

18 Aug 2021

We are grateful to the reviewers for their suggestions which greatly help improve the article, achieving journal quality.

Reviewer #1: The paper presents interesting an unique study worth publication. However, there are some issues that need correction.

1. English language should be checked and corrected by professional English editor.

Answer: Thank you very much for the suggestion as it improved the article. English has been reviewed by a professional English editor.

2. The introduction is too short- it should more deeply justify the need of the study and give more scientific background.

Answer: Thank you very much for the suggestion as it improved the article. We have modified the introduction to better justify the need of the study and providing more scientific backgrounds.

3. Please underline the novelty of the study in the introduction.

Answer: Thank you very much for the suggestion as it improved the article. I improved the novelty of the study in the introduction.

4. Justify the childrens' age. Why you have included children in the age of 10-11??

Answer: Thank you very much for the suggestion as it improved the article. We decided to include children aged 10-11 years because these children currently participate in an ongoing larger cohort study project and we included the justification in the article.

5. The total population calculated for the study was 316; however you managed to include only 73. This is great weak point of the study and should be pointed broadly in limitation. In title please point that this is a pilot study or preliminary study- the complete study should include the total of 316 children.

Answer: Thank you very much for the suggestion as it improved the article. I added as a limitation that the total sample of the study should be 316 but only 73 individuals were included. We modified the title pointing out that this is a preliminary study.

6. Provide the manufacturer of digitale scale. Was the body measured fasting?

Answer: Thank you very much for the suggestion as it improved the article. We provided the information on the digital scale manufacturer. Bodies were measured fasting, and we added this information in the article.

7. In methods please clearly provide information on what kinds of "events that occurred in the gestation period" were analyzed in the study.

Answer: Thank you very much for the suggestion as it improved the article. In the methods section we provided information about which types of events that occurred during the gestation period were analyzed.

8. The RESULT section should start from the sentence "The results obtained demonstrate that there was a significant correlation between the number of ridges on the child’s fingers, ... "atline 216.

Answer: Thank you very much for the suggestion as it improved the article. We started the results section with the phrase "The results obtained demonstrate that there was a significant correlation with the number of ridges on the child's fingers".

9. In table headings you use AGM abbrev., but in the table there is IGM- correct it.

Answer: Thank you very much for the suggestion as it improved the article. We made the correction requested.

10. In results (text and tables) clearly state whose BMI do you mean- child or maternal.

Answer: Thank you very much for the suggestion as it improved the article. We mean child's BMI and we made it clearer in the text.

11. "Different genetic variants have been identified as monogenic forms of human obesity, having success over common polygenic forms." This sentence is not clear. Do you mean that monogenic forms of obesity predominate over polygenic? It is not true- correct it.

Answer: Thank you very much for the suggestion as it improved the article. We modified the sentence, making it clearer and we added one more article as a reference for better support.

12. Your discussion is a simple enumeration of the results of similar studies. Please provide some molecular/genetic mechanisms linking obesity and fingerprints, some clinical implications and directions of future studies in the issue.

Answer: Thank you very much for the suggestion as it improved the article. We have provided genetic aspects linking obesity and fingerprints in the discussion section on page 18 and added clinical implications and directions for future studies in the discussion on page 20.

13. In the study aim and title you pointed, that you want to analyze the association between events occurring in the gestational period and the occurrence of obesity in children from dermatoglyphic traits. However, in the study you did not present any results relating to "events occurring in the gestational period". Namely, you did not investigate events occurring in the gestational period. The only "event" analyzed was maternal age at gestation onset. Please correct study title and aim deleting "events occurring in the gestational period". Unfortunately, the lack of the analysis of "events occurring in the gestational period" significantly decreases the scientific value of the paper.

Answer: Thank you very much for the suggestion as it improved the article. We excluded from the title "events occurred during the gestational period", modifying by “Association between gestational age”.

14. After limitations please add some study strong points.

Answer: Thank you very much for the suggestion as it improved the article. After limitations, we added the strong points of the study.

15. "The study shows the application of research in dermatoglyphics as an epigenetic predictive obesity trait, and the results show predictive obesity traits in children. In addition, the results were associated with gestational data." Such sentence should be placed in the discussion section. Moreover, you did not discussed any "epigenetic" link between obesity and fingerprint. Thus, the use of word "epigenetic" in the paper is unjustifiable.

Answer: : Thank you very much for the suggestion as it improved the article. We added to the discussion the sentence “The study shows the application of research in dermatoglyphics as an epigenetic predictive obesity trait, and the results show predictive obesity traits in children and I added epigenetic information in the introduction and discussion of the article.

Reviewer #2: The authors have presented useful data

I have the following comments

1) The authors have used the term predictive traits; however, they have just used correlations. Even regressions have not been used for analysis

Answer: We are grateful for your important suggestion. Actually, a multivariate approach may be appropriate in this type of study. However, we tested all assumptions for an exploratory analysis and our data violated some important requirements for the correct interpretation of MANOVA and MANCOVA. This was observed by the presence of multicollinearity and homoscedasticity. In fact, we performed a multivariate analysis trying to control these assumptions, but the error was too high to prevent our careful interpretation of the true results. Even the small sample in our study did not allow for a cautious interpretation of the multivariate analysis. Neither the data, nor even the size of our sample (to adopt significance at p<0.05), were sufficient to meet the assumptions of the multivariate analysis, which ended up being violated. The multiple regression followed by stepwise backward analysis presented in the study was an attempt to deal with it in a multivariate way, with predictors of building less complex models to interpret and control (for our purpose). The univariate analysis was then carried out; despite the presence of several statistical tests (such as the chi-square and Spearman correlation), each effect could be analyzed with less error, allowing for a more adequate and cautious interpretation, followed by a vision more holistic of the results given by linking our data.

2) The results section merges into methods. It is difficult to understand the whole results section

Answer: Thank you very much for the suggestion as it improved the article. We separated the methods and results sections for a better understanding 

3) Do we have information on the mothers? Nothing much has been included in the manuscript

Answer: Thank you very much for the suggestion as it improved the article. The data about the mothers that we have are the gestational data that we better explained on page 9.

Reviewer #3: Here the authors analyze the association between events during gestation and the occurrence of obesity in children from dermatoglyphic traits. The authors report a predictive trait of obesity when comparing BMI and fingerprint groups in the figure patterns for males.

1) This is a preliminary correlation study and does not have any novel mechanistic insight.

Answer: Thank you very much for the suggestion as it improved the article. We added in the title that it is a preliminary unpublished study.

2) The sample size (73) is too small to have any predictive value.

Answer: Thank you very much for the suggestion. Yes, the sample of 73 individuals is small indeed, and it was jeopardized due to the COVID-19 pandemic, but the study and data processing were carried out carefully so as not to violate the assumptions. Despite the sample being small, the result is consistent, in addition to being an unprecedented study.

---

## [Decision Letter · Decision Letter 1]

25 Aug 2021

Association among events that occurred in the gestation period and obesity in children with the use of dermatoglyphic traits

PONE-D-21-09898R1

Dear Dr. Alberti,

We’re pleased to inform you that your manuscript has been judged scientifically suitable for publication and will be formally accepted for publication once it meets all outstanding technical requirements.

Kind regards,

Ramune Jacobsen

Academic Editor

PLOS ONE

Reviewers' comments:

**Comments to the Author**

1. If the authors have adequately addressed your comments raised in a previous round of review and you feel that this manuscript is now acceptable for publication, you may indicate that here to bypass the “Comments to the Author” section, enter your conflict of interest statement in the “Confidential to Editor” section, and submit your "Accept" recommendation.

Reviewer #1: All comments have been addressed

Reviewer #2: All comments have been addressed

2. Is the manuscript technically sound, and do the data support the conclusions?

Reviewer #1: Yes

Reviewer #2: Yes

3. Has the statistical analysis been performed appropriately and rigorously? 

Reviewer #1: I Don't Know

Reviewer #2: Yes

4. Have the authors made all data underlying the findings in their manuscript fully available?

Reviewer #1: Yes

Reviewer #2: Yes

5. Is the manuscript presented in an intelligible fashion and written in standard English?

Reviewer #1: Yes

Reviewer #2: Yes

6. Review Comments to the Author

Reviewer #1: All comments have been addressed.

All changes have been implemented. I recommend accepting the manuscript.

Reviewer #2: The authors have addressed my queries. They have justified their reasons for using the methods. It may be appropriate if these add these details in the manuscript as well.

7. PLOS authors have the option to publish the peer review history of their article (what does this mean?). If published, this will include your full peer review and any attached files.

Reviewer #1: No

Reviewer #2: No

---

## [Editor Report · Acceptance letter]

27 Aug 2021

PONE-D-21-09898R1 

Association between gestational period and obesity in children with the use of dermatoglyphic traits: A preliminary study 

Dear Dr. Alberti:

I'm pleased to inform you that your manuscript has been deemed suitable for publication in PLOS ONE. Congratulations! Your manuscript is now with our production department. 

Kind regards, 

on behalf of

Dr. Ramune Jacobsen 

Academic Editor

PLOS ONE